# CReTIHC: Designing Causal Reasoning Tasks about Temporal Interventions and Hallucinated Confoundings

**Changwoo Chun**[1], **Songeun Lee**[1], **Jaehyung Seo**[2] **and Heuiseok Lim**[2†]
[1]Hyundai Motor Company
[2]Department of Computer Science and Engineering, Korea University
[1]{cwchun,songeun.lee}@hyundai.com
[2]{seojae777,limhseok}@korea.ac.kr

## Abstract

Large language models (LLMs) have demonstrated impressive capabilities in natural language processing. However, their ability to establish causal relationships, particularly in the context of temporal interventions and language hallucinations, remains challenging. This paper presents **CReTIHC**, a novel dataset designed to test and enhance the causal reasoning abilities of LLMs. The dataset is constructed using a unique approach that incorporates elements of verbal hallucinations and temporal interventions through the reengineering of existing causal inference datasets. This transformation creates complex scenarios that push LLMs to critically evaluate the information presented and identify cause-and-effect relationships. The CReTIHC dataset serves as a pioneering tool for improving LLM's causal inference capabilities, paving the way for a more nuanced understanding of causal relationships in natural language processing (NLP) tasks. The whole dataset is publicly accessible at: (https://github.com/ChangwooChun/CReTIHC)

## 1 Introduction

LLMs have emerged as a powerful tool in natural language processing, with the ability to generate human-like text by exploiting extensive prior knowledge. Despite these advancements, LLMs are based on the Transformer structure (Vaswani et al., 2017) and have exhibited shortcomings in establishing causal relationships in their outputs. This stems from their reliance on empirical reasoning and statistical patterns from training data (Zhao et al., 2023). Specifically, LLMs face challenges in disentangling true causal relationships from confounding biases introduced by language hallucinations and temporal interventions (Zhang et al., 2022).

In this study, we introduce a novel set of tasks and an accompanying dataset, **CReTIHC** (**C**ausal **Re**asoning tasks about **T**emporal **I**nterventions and **H**allucinated **C**onfoundings), specifically designed to assess and enhance the causal reasoning capabilities of LLMs. Our tasks are rooted in commonsense reasoning and involve a sequence of events that must be analyzed temporally to ascertain cause-and-effect relationships. By integrating elements of language hallucinations and temporal interventions, we present LLMs with challenging scenarios that require critical evaluation of information.

The CReTIHC dataset was derived by redesigning an existing natural language commonsense causal reasoning dataset, adding complexity by exploiting the difference between temporal and causal order and correlation and causation of word co-occurrences. This made the task of causal reasoning easy for humans, but LLMs based on word co-occurrence probabilities had difficulty clearly understanding causality from noise caused by temporal intervention and confounders in the form of chains, forks, and colliders commonly addressed in existing causality studies (Guo et al., 2020).

In our evaluation, we delve into the dynamic nature of causal relationships in text-based languages and their susceptibility to influences from temporal cues and sequences. Our findings indicate that LLMs require further refinement in their causal inference capabilities, particularly when utilizing the CReTIHC dataset. This underscores the necessity for a more fundamental approach to understanding causality within natural language.

The integration of LLMs into the dataset construction process presents a significant advancement. This not only enables the generation of consistent data based on given instructions but also reduces the cost and labor associated with traditional crowdsourcing methods. This innovative methodology paves the way for the creation of larger, more intricate datasets, demonstrating that the thought-

---
† Corresponding author

ful motivations and innovative ideas of NLP researchers can lead to the generation of diverse and complex NLP task datasets.

This research is instrumental in addressing the limitations of LLMs, thereby contributing to the development of more robust and causal-aware natural language processing systems.

## 2 Related Works

Causal reasoning, the process of identifying and understanding the cause of certain outcomes and the influence of changes in these causes, has been a long-standing focus in various fields (RA, 1958; Cochran and Chambers, 1965; Rosenbaum, 2022). In the domain of deep learning-based natural language processing, efforts have been made to incorporate and interpret causal reasoning to discern how causal factors contribute to the outcomes generated by language models (Wood-Doughty et al., 2018; Sridhar and Getoor, 2019; Veitch et al., 2020; Keith et al., 2020; Feder et al., 2021).

In the era of deep learning, causal reasoning has emerged as a crucial capability to ensure that models generalize well and produce accurate outcomes when faced with new data that may not conform to the training data distribution (Schölkopf et al., 2021). Although the scaling laws have facilitated the emergence of models with human-like generalization capabilities (Radford et al., 2018; Brown et al., 2020; Chowdhery et al., 2022), such as GPT-3, GPT-4[1], and LLaMA (Touvron et al., 2023), by leveraging extensive data and large architectures, these models often struggle to establish causality. Pre-training on extensive data does not suffice in ensuring causal inference as causal relationships are often obscured by interventions, confounding biases, and language hallucinations (Feder et al., 2022). Recently, to enhance causal reasoning in models, methods such as adversarial training and data augmentation techniques rooted in linguistic theories of causal connectives have been introduced (Staliūnaitė et al., 2021). Notably, these methods have achieved statistically significant improvements on the COPA dataset, including its balanced version designed to avoid surface-level cues.

Recently, there have been efforts in creating tasks, frameworks, and datasets focusing on commonsense causal reasoning, such as COPA (Gordon et al., 2012), αNLI (Bhagavatula et al., 2020),

ROCK (Zhang et al., 2022), and e-CARE (Du et al., 2022). These aim at assessing causal inference abilities independently of the sheer volume of knowledge. However, existing datasets and tasks have limitations in measuring and evaluating the dynamic nature of causal relationships, especially in scenarios involving sequential events leading to a particular outcome. Furthermore, these datasets don't address confounding biases introduced through temporal signals or language hallucinations stemming from word co-occurrences

To address these limitations, our study restructures the e-CARE[2] dataset to present tasks centered around temporal interventions and hallucinated confoundings. These tasks are designed to be more challenging and representative of real-world scenarios. This approach aims to provide a foundation for more robust and fine-grained analysis of causality in natural language processing, particularly in the era following the advent of LLMs.

## 3 Proposed Method

Our key idea is to investigate changes in the causal relationship between two events when a new event appears. We designed the CReTIHC dataset by adding temporal interventions and hallucinated confoundings to the e-CARE, an existing natural language commonsense causal reasoning dataset. Grounding prior studies regarding causality (Imbens and Rubin, 2015; Keith et al., 2020; Zhang et al., 2022; Feder et al., 2022), we have noticed that unseen, unobserved interventions can heavily impact the causal relationship between events. Also, previous causality research shows that causal relationships guarantee antecedent relationships, but antecedent relationships do not guarantee causal relationships (Russell, 1912; Bunge, 2017; Zhang et al., 2022).

To examine whether LLMs can identify temporal order and actual causal relationships, both temporal order relationships confused with actual causal relationships are needed. However, the original e-CARE dataset, which is not arranged chronologically, consists of premises, hypotheses, and explanations that make the hypotheses plausible. As described in Figure 1, we restructured this dataset to create a chronologically ordered sequence of events. When the premise and hypothesis are in an *effect* relationship, we reordered them so that the premise is the event that occurred first and the

---

[1]OpenAI. (2023). ChatGPT & GPT-4 https://openai.com

[2]https://github.com/waste-wood/e-care

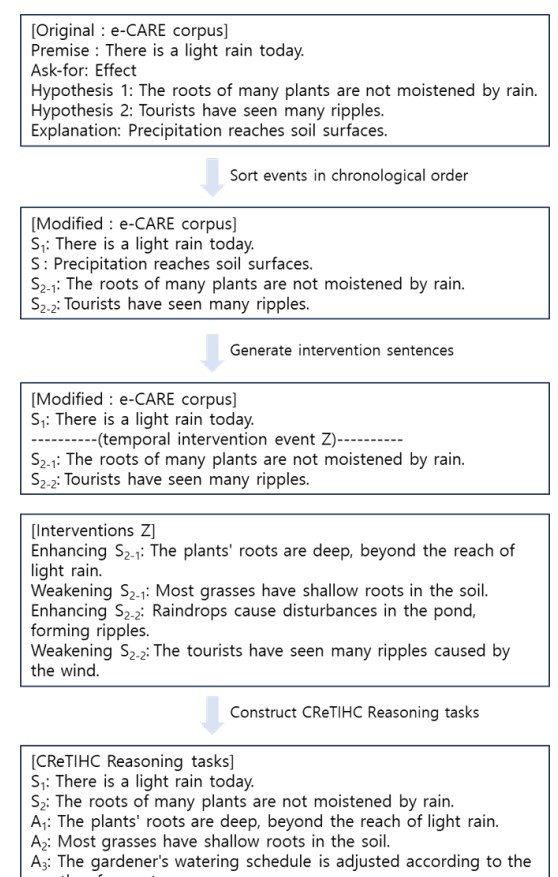

[Original : e-CARE corpus]
Premise : There is a light rain today.
Ask-for: Effect
Hypothesis 1: The roots of many plants are not moistened by rain.
Hypothesis 2: Tourists have seen many ripples.
Explanation: Precipitation reaches soil surfaces.

*Sort events in chronological order*

[Modified : e-CARE corpus]
$S_1$: There is a light rain today.
$S$: Precipitation reaches soil surfaces.
$S_{2-1}$: The roots of many plants are not moistened by rain.
$S_{2-2}$: Tourists have seen many ripples.

*Generate intervention sentences*

[Modified : e-CARE corpus]
$S_1$: There is a light rain today.
----------(temporal intervention event Z)----------
$S_{2-1}$: The roots of many plants are not moistened by rain.
$S_{2-2}$: Tourists have seen many ripples.

[Interventions Z]
Enhancing $S_{2-1}$: The plants' roots are deep, beyond the reach of light rain.
Weakening $S_{2-1}$: Most grasses have shallow roots in the soil.
Enhancing $S_{2-2}$: Raindrops cause disturbances in the pond, forming ripples.
Weakening $S_{2-2}$: The tourists have seen many ripples caused by the wind.

*Construct CReTIHC Reasoning tasks*

[CReTIHC Reasoning tasks]
$S_1$: There is a light rain today.
$S_2$: The roots of many plants are not moistened by rain.
$A_1$: The plants' roots are deep, beyond the reach of light rain.
$A_2$: Most grasses have shallow roots in the soil.
$A_3$: The gardener's watering schedule is adjusted according to the weather forecast.

Figure 1: CReTIHC: Changing the causal relationship of the e-CARE dataset

hypothesis is the event that occurred second. Conversely, when the premise and hypothesis are in a *cause* relationship, we reordered them so that the hypothesis is the event that occurred first and the premise is the event that occurred second. The preceding event is denoted as $S_1$, and the succeeding event is denoted as $S_2$.

In the next phase, we employed LLMs to manipulate the causal relationships in the restructured dataset, presented as a series of consecutive events. We used temporal interventions (**Task 1**) and introduced confounders through linguistic manipulation (**Task 2**), with the LLM generating sentences that vary the causal relationship between $S_1$ and $S_2$. Hallucinated confounders, adding new elements to make spurious causal relationships, were utilized to create confusion between word co-occurrence and actual causality. An event that changes the causal relationship between $S_1$ and $S_2$ are denoted by $Z$.

**Task 1: Temporal Interventions**  Task 1 uses a temporal intervention to manipulate the causal relationship between event $S_1$ and outcome $S_2$ based on the timing of the causal event $S_1$. If $S_1$ and $S_2$ have a plausible causal relationship, we intervene with an event that could occur between $S_1$ and $S_2$ to weaken the causal relationship. If $S_1$ and $S_2$ do not have a causal relationship with each other, an intervention can strengthen the causal relationship between the two events. The event $Z$ is positioned between the two events, forming a chain that alters the relationship between the causally related events. Depending on its temporal placement, an intervention can relatively transform the strength of causality between the two events $S_1$ and $S_2$.

**Prompts of Task 1**  As described in Table 1, we consistently used the directive in all prompts: `"Never use words or phrases that have already been used, and keep them short and concise, no more than 10 words."` This was implemented to mitigate the tendency of the LLMs to generate wordy sentences and repetitively use previously mentioned words or phrases. The aim was to guide the LLMs towards generating more concise and diverse expressions. Then, two verbs were used to create each sentence with the causal relationship changing in the same direction (e.g. 'make' and 'enhance' to strengthen causal relationships). This reason is based on the experimental observation that it is an ideal way to choose the one that produces better sentences between making causal relationships and strengthening them according to the sample.

| Direction | Prompt |
|---|---|
| **Weaken** | Write a sentence that weakens the causal relationship between $S_1$ and $S_2$. |
| **Eliminate** | Write a sentence that completely eliminates the causal relationship between $S_1$ and $S_2$. |
| **Make** | Write a sentence that makes the causal relationship between $S_1$ and $S_2$. |
| **Enhance** | Write a sentence that enhances the causal relationship between $S_1$ and $S_2$. |

Table 1: A prompt to create a temporal intervention that transforms the causal relationship between S1 and $S_2$

**Task 2: Hallucinated Confoundings**  Task 2 examines cases where no causal relationship exists between event $S_1$ and $S_2$, but a confounding event z obscures this fact. The task involves two structures: a Collider and a Fork. In the collider structure, the confusing event z is causally related to both $S_2$ and

$S_1$. Here, no direct causal relationship exists between $S_2$ and $S_1$, but event $S_1$ is changed to imply a causal relationship. In the fork structure, both $S_2$ and $S_1$ are the result of the confusion event z, and event $S_1$ is changed to imply a causal relationship that does not exist. This creates confusion that there is a causal relationship between $S_1$ and $S_2$, even though there is no direct causal relationship between them.

**Prompts of Task 2**  As shown in Table 2, to integrate Hallucinated Confounding events, we crafted sentences encompassing confounders, specifically Forks and Colliders, drawing upon the principles of Structural Causal Models (SCMs) (Guo et al., 2020). These SCMs exert influence on causality, thereby enabling us to seamlessly introduce confounding elements into our narrative. Notably, to ensure the effectiveness of the confounding, we permitted the reuse of words.

| Confounding | Prompt |
|---|---|
| **Fork** | Write a sentence with a new factor that affects both S1 and S2 and confounds the causal relationship. |
| **Collider** | Write a sentence with a new factor that is influenced by both S1 and S2 and confounds the causal relationship. |

Table 2: A prompt that creates confounding factors in the causal relationship between S1 and S2

Task 1 and Task 2 highlight the importance of context in interpreting causality, demonstrating how linguistically introduced confounders and temporal interventions can alter the perception of causal relationships. The fundamental premise of the CReTIHC is maintaining the arrangement of sentences according to the temporal order of events, providing a more challenging and realistic task for evaluating the causal reasoning abilities of LLMs.

## 3.1 Self-refining Technique

We needed a process to critically evaluate and revise sentences generated by LLM to obtain a high-quality causal reasoning dataset. Inspired by the self-refinement method (Madaan et al., 2023), we integrated self-refining the sentences generated by LLM and rewriting inappropriate sentences into the data collection process.

The self-feedback step determines whether the generated sentences fit the given instructions and guidelines. If conditions are not met or errors are found, in the self-refining phase, LLM autonomously rectifies the sentences to suit the direction of the task better. At this time, more fine-grained sentences can be obtained using the previously generated results as input to the LLM, along with an additional prompt and few-shot examples.

This self-refining process is iterative and LLM continually evaluates and improves its results until it reaches a satisfactory level of accuracy. The reason for not providing all information as input from the beginning is based on our experimental experience that sentences obtained through repeated operations of inputting the output of LLM are of higher quality than sentences obtained by providing a lot of information from the beginning.

Incorporating these self-refining techniques into your data collection methods will improve the quality of the text produced. On the other hand, due to lack of time and cost considerations, we only repeated this iteration a maximum of three times. If the final results did not show sufficiently satisfactory results in the self-feedback phase even after three iterations, they were excluded from the CReTIHC dataset.

**Prompts of Self-refining**  Self-refining process is divided into two steps: Self-feedback and self-refining. In the self-feedback step, the model evaluates a generated sentence that follows a sequence of events, S1 and S2. In the self-refining step, the model is prompted to rewrite the sentence by introducing a new confounding factor that either eliminates or establishes a causal relationship between S1 and S2. This step is executed when the quality of the sentence generated through the previous self-feedback is low. Sentences that have gone through self-refining are used as input to the self-feedback step again iteratively. An example of a prompt used for self-refining is in Appendix B.

## 4  Experiments

Our proposed CReTIHC dataset introduces the task of identifying all three factors (Enhance, Weaken, Irrelevance) involved in a causal relationship. When appearing an event between two chronologically ordered events, S1 and S2, it is a matter of determining how that event affects the existing causal relationship between S1 and S2. A total of 5 lines of sentences are given in one sample; the S1 and S2 events are fixed, and the task is to determine how each of the three sentences, A1 to A3, affects the causal relationship between the previous

|          | Accuracy | F1    |       |       |
|----------|----------|-------|-------|-------|
|          |          | True  | False | None  |
| **GPT-3**    | 0.671    | 0.736 | 0.733 | 0.501 |
| **ChatGPT**  | 0.527    | 0.664 | 0.539 | 0.331 |

Table 3: Quantitative evaluation results of LLMs

two events. To exclude the influence of sentence order of prompts, the positions of Enhance, Weaken, and Irrelevance were randomly mixed.

- **True**: *Enhancing* a causal relationship between $S_1$ and $S_2$.
- **False**: *Weakening* a causal relationship between $S_1$ and $S_2$.
- **None**: *Irrelevance* = Not changing a causal relationship between $S_1$ and $S_2$.

**LLM Evaluation** In our experiments using 1,000 sentences from the CReTIHC dataset, even advanced LLMs like GPT-3 and GPT-3.5 struggled to accurately identify causal relationships, as shown in Table 3 In particular, the accuracy of determining whether the causal relationship has been weakened or has no effect on the causal relationship is lower than the accuracy of determining whether the causal relationship has been strengthened. These results support the hypothesis that language models trained on word occurrence probabilities have difficulties distinguishing whether co-occurring words have a causal relationship, especially in the presence of temporal interventions and hallucinated confounding. These results highlight the potential of the CReTIHC dataset in causal inference. Our dataset allows us to measure the strength of causal relationships using ranking methods, opening new avenues for research in this field. The detailed version and parameters of the model used in the evaluation are in the Appendix A.

**Human Evaluation** We conducted a human evaluation of 100 samples from the CReTIHC dataset. Four evaluators were provided with the same prompts as those used in the LLM, and they were tasked with assessing causality between sentences, with labels True, False, and None. The evaluators weren't informed about the actual labels but were asked to identify samples with causal inference errors. The human evaluators achieved an average accuracy of 0.895 on the CReTIHC dataset. They excelled at identifying strengthened causal relationships but were less accurate with weakened or irrelevant ones. The Table 4 summarizes the

results from a sampled set of 100 problems (a total of 300 causal reasoning assessments):

| Evaluator | TRUE | FALSE | None  |
|-----------|------|-------|-------|
| Human A   | 0.96 | 0.87  | 0.81  |
| Human B   | 0.92 | 0.91  | 0.89  |
| Human C   | 0.92 | 0.85  | 0.88  |
| Human D   | 0.97 | 0.89  | 0.88  |
| Average   | 0.94 | 0.88  | 0.865 |

Table 4: Experiments with human evaluators

Post-experiment feedback emphasizes that external knowledge or personal experience sometimes influences judgment. There was some confusion among some evaluators about the distinction between weakened and nonexistent causal relationships, resulting in 2.3% of the sentences being rejected. To measure the consistency of agreement between raters, Fleiss' kappa coefficient[3] was calculated, resulting in a score of 0.7579. This suggests that there was considerable agreement between the raters.

## 5 Conclusions

The significance of causal reasoning in language understanding cannot be overstated. Our research has sought to reflect the complexity of real-world causal reasoning by enhancing an existing commonsense causal reasoning dataset with temporal interventions and hallucinated confoundings. The CReTIHC dataset, rich with continuous event sequences and confounding variables, can offer a more rigorous and holistic exploration of causal relationships in language. Our approach mirrors real-world causal reasoning scenarios and has leveraged LLMs to generate comprehensive datasets. This underscores the potential of LLMs in enhancing various tasks in NLP and highlights the need for models that can efficiently learn from small amounts of data and understand causal relationships as adeptly as humans do. While we acknowledge that a language model trained on our dataset may not fully resolve causal inferences, we believe that our work with the CReTIHC benchmark performance measurements contributes to the ongoing journey toward full causal understanding by machines. In conclusion, our research underscores the importance of causal reasoning in language understanding and paves the way for future research in this critical area.

---

[3] https://en.wikipedia.org/wiki/Fleiss%27_kappa

## Limitations

While our study presents an innovative approach to augmenting the causal reasoning capabilities of LLMs, it is important to acknowledge its inherent limitations.

Firstly, our dataset is derived from the e-CARE dataset, which may limit scenario diversity and potentially contain biases or typos from the source data.

Secondly, the self-refinement technique embedded in our methodology, although beneficial in enhancing sentence quality, is not infallible and may not consistently yield optimal refinements. Due to resource constraints, we limited the iterative process to a maximum of three rounds.

Thirdly, we assessed GPT-4's performance and achieved a high accuracy of 0.92. However, we suspect this result may stem from circular evaluation, where the model is assessed using data it generated rather than truly reflecting GPT-4's inference capabilities. This raises questions about the suitability of our dataset for evaluating GPT-4.

Lastly, our study primarily focuses on tasks in the English language. The generalizability of our findings to other languages remains an open question.

Going forward, our objectives include expanding the CReTIHC dataset by incorporating a broader spectrum of causal scenarios. We also plan to develop more sophisticated self-refinement techniques capable of iteratively adjusting sentence quality and difficulty beyond current limitations. Furthermore, we remain open to post-release collaboration with human annotators to address any additional errors or issues that may arise after dataset publication.

## Ethics Statement

Our research uses LLMs and does not directly involve human subjects or personal data. The development and use of LLMs can have serious social implications, including the potential for misuse to create misleading or harmful content. To alleviate these concerns, we have focused and worked on improving the causal inference capabilities of LLM, which can contribute to more accurate and reliable results. We also inspected the data for any social bias or harmful expressions that may have been included. Our dataset, CReTIHC, is publicly available and can be used by other researchers to further enhance the capabilities of LLM. We encourage the responsible use of our research findings and datasets.

## Acknowledgments

This research was supported by Basic Science Research Program through the National Research Foundation of Korea (NRF) funded by the Ministry of Education (NRF-2021R1A6A1A03045425).

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

## A  Details of the Language Models

For creating sentences involving temporal interventions and hallucinated confoundings, we utilized GPT-4 due to its advanced capabilities in understanding and manipulating causal relationships. However, for self-checking processes, we used ChatGPT. This combination of models allowed us to effectively generate and refine the dataset, leveraging the strengths of each model.

To evaluate the causal inference performance of the LLM, zero-shot performance was measured on the dataset. Specifically, the text-davinci-003 model based on GPT-3, ChatGPT (gpt-3.5-turbo),

and GPT-4 model were used for this purpose. The API and all parameters used default values except those shown in the Table 5.

| Type | Version & Value |
|---|---|
| GPT-3 [4] | text-davinci-003 |
| ChatGPT [5] | gpt-3.5-turbo-0301 |
| GPT-4 [6] | gpt-4-0314 |
| temperature | 0.0 |
| top_p | null |

Table 5: Details of the ChatGPT Model

## B  Prompts of Self-refining

| Step | Prompt |
|---|---|
| **Self-Feedback** | Answer the sentence given after S2 based on the conditions below. Response 'True' if all of the conditions below are met, and 'False' if any of the following condition is not met: Condition 1. Does a sentence *enhance/weaken* the causal relationship between S1 and S2? Condition 2. Is a sentence an event located between S1 and S2? |
| **Self-Refining** | For the given sentence after S2, rewrite the sentence by adding a new confounding factor to *make/eliminate/confuse* the causal relationship between S1 and S2. Never use words or phrases that have already been used, and keep them short no more than 10 words. |

Table 6: Prompt to self-feedback whether the generated sentence meets the appropriate conditions and correct and select the sentence

## C  Prompts for Experiments

In our experiments, we utilized the following prompt for the LLMs:

"Here are five sentences: S1, S2, A1, A2, and A3. S1 and S2 depict sequential events. Your task is to determine which among A1, A2, and A3 undermines the causal relationship between S1 and S2, and which one strengthens it. Remember that A1, A2, or A3 might be unassociated with S1 and S2. You should pinpoint one sentence from A1 to A3 that amplifies the link between S1 and S2, and another sentence from A1 to A3 that impedes this connection.

Answer 'True' if A# strengthens the causal relation between S1 and S2. Answer 'False' if A# weakens causality between S1 and S2. Answer 'None' if A# clearly does not affect the causation between S1 and S2.

Please assess A1, A2, and A3 individually based on their relevance to S1 and S2. Don't be verbose. Only return results among "True", "False" or "None". Do not generate any further explanation."

This prompt was designed to encourage the LLM to critically evaluate the relationship between the given sentences and make a determination about the nature of the causal relationship. We've used some paraphrasing so that the prompts don't overlap too closely with the sentences for creating the dataset.

## D  Details of CReTIHC dataset

From the e-CARE training set, a total of 14.9K sets of initial datasets were collected for causality evaluation. Of these, about 47% were filtered through the self-refining process, and as a result of additional filtering through the final manual work of annotators, the CReTIHC dataset consisted of 2,638 causal inference test sets. The criteria provided as a filtering guide for annotators are:

1. In the case of a sentence that refers the causal relationship by directly using the sentence number '$S_1$' or '$S_2$' (e.g "$S_1$ and $S_2$ do not have a direct relationship.")

2. In case of multiple sentences instead of a single sentence

3. In case of one instance set is incompletely created (if any of S1, S2, Enhance, Weaken, or Irrelevance is missing): Most of these cases are caused of OpenAI API error or usage quota limit.

4. In case of sentences rejected by human annotators due to broken text encoding or inappropriate.

Statistics of the total data are in Table 7.

We also rectified errors in the e-CARE dataset. Although LLM is robust to small typos, in subtle tasks such as causal inference, even minor differences in words could have a large impact on causal

| Basic Statistics | CReTIHC |
|---|---|
| # of Test-case | 2,638 |
| # of Total Sentences | 14,565 |
| # of Unique Sentences | 13,324 |
| - Average length | 52.6 |
| # of Words | 126K |
| # of Unique Words | 11K |

Table 7: Statistics for CReTIHC dataset

relationships. For example, an event that has an effect on Tomy may not have an effect on Tony. Even errors that could be overlooked by humans were a major drawback of the CReTIHC dataset generated from LLM. Therefore, we modified the original data to ensure accuracy and consistency. Of the 14,028 sentences in our dataset that directly corresponded to e-CARE, 976 were meticulously edited to address spelling errors.

## E    Samples of CReTIHC dataset

The CReTIHC dataset consists of six columns. The first column, 'IDX', denotes the individual instance index within the CReTIHC dataset. The second and third columns, '$S_1$' and '$S_2$', represent two sequential events from e-CARE. The fourth column, 'TRUE', contains events that strengthen the causal relationship between $S_1$ and $S_2$. The fifth column, 'FALSE', includes events that weaken this causal relationship. The sixth column, 'NONE', introduces confounding factors that add ambiguity to the causal relationship between $S_1$ and $S_2$.

| IDX | S1 | S2 | TRUE | FALSE | None |
|---|---|---|---|---|---|
| 1 | There is a light rain today. Precipitation reaches soil surfaces. | The roots of many plants are not moistened by rain. | The plants' roots are deep, beyond the reach of light rain. | The roots of the majority of plants are very shallow. | The gardener's watering schedule is adjusted according to the weather forecast. |
| 2 | Susan wants to buy a restricted pesticide. Rotenone is a restricted-use pesticide. | She bought rotenone. | She was granted permission to purchase restricted-use pesticides. | Rotenone's use restrictions were recently lifted. | Local farming communities rigorously review permits to purchase rotenone pesticides. |
| 3 | He greeted the orcas in the water. Orcas are very social animals. | The orcas spouted water to respond to him. | TThe man is good at imitating the sounds of killer whales. | The orcas were simply expelling water as part of their breathing. | A sudden wave crashed, startling both the man and the orcas. |
| 4 | Tom eats a lot of eggs every day. | His hatchery can produce hundreds of millions of fertilized eggs every year. | Tom's high egg consumption fuels his hatchery's production. | Tom's egg consumption is unrelated to his hatchery's output. | Gallstones are lumps composed mainly of cholesterol. |
| 5 | Tom eats a lot of eggs every day. Gallstones are lumps composed mainly of cholesterol. | The cholesterol content in his body is extremely high, so he suffers from gallstones. | He's gallstones are a direct result of his dietary choices. | Tom is also taking medication to lower his cholesterol levels. | Jack's doctor pointed out his habit of eating a lot of eggs. |
| ... | ... | ... | ... | ... | ... |

Table 8: Examples of CReTIHC dataset