# OpenReview forum: "CReTIHC: Designing Causal Reasoning Tasks about Temporal Interventions and Hallucinated Confoundings"
_EMNLP/2023/Conference — EMNLP 2023 Findings_

### Official Review · Reviewer_eRnj · 2023-07-26

**Soundness:** 3

**Excitement:**

4: Strong: This paper deepens the understanding of some phenomenon or lowers the barriers to an existing research direction.

**Missing References:**

"Improving commonsense causal reasoning by adversarial training and data augmentation" by I Staliunaite, PJ Gorinski, I Iacobacci (2021) have presented a very similar method of generating confounders as well as generating adversarial examples to improve commonsense causal reasoning models.

**Paper Topic And Main Contributions:**

The paper presents a causal reasoning dataset, which has been constructed from already existing causal inference datasets, by using verbal hallucinations and temporal interventions on the existing data points. The new dataset presents the task of not only predicting the presence of a causal link between clauses, but also the prediction of whether another clause strengthens or weakens this relationship.

**Questions For The Authors:**

Line 257: The method described in this paragraph sounds like adversarial training, is it different in some way or are there other reasons why you do not use this term to describe it?


**Reasons To Accept:**

A very interesting way of combining causal inference theory on forks, chains and colliders for training better commonsense reasoning models.


**Reasons To Reject:**

Line 169: The main part of the data collection is placed in the appendix. This should have definitely been in the main body of the paper.


**Reproducibility:**

2: Would be hard pressed to reproduce the results. The contribution depends on data that are simply not available outside the author's institution or consortium; not enough details are provided.

**Reviewer Confidence:**

4: Quite sure. I tried to check the important points carefully. It's unlikely, though conceivable, that I missed something that should affect my ratings.

**Typos Grammar Style And Presentation Improvements:**

Line 204: Very repetitive explanation.

Line 300: What does ‘reversely weaken’ mean? Is this meant to be described in contrast to strengthening? Better phrasing could help make it more clear.

---

> ### Author Rebuttal · Authors · 2023-08-29
>
> Thank you for your insightful comments. We appreciate the time and effort you've invested in reviewing our work. We would like to address the concerns you raised:
>
> > R1 : *‘Line 169: The main part of the data collection is placed in the appendix. This should have definitely been in the main body of the paper.’*
>
> **[Data Collection in the Appendix]**
>
> We acknowledge your concern regarding the placement of the main part of the data collection in the appendix. Given the page limit of 4 for short papers, we made this decision. However, if our paper is accepted, we will move this section to the main body in the camera-ready version, which allows up to 5 pages.
>
> > R2 : *‘Line 257: The method described in this paragraph sounds like adversarial training, is it different in some way or are there other reasons why you do not use this term to describe it?’*
>
> **[Differences from adversarial training]**
>
> In response to your question about the similarity of our method to adversarial training: While both approaches aim to challenge and improve model performance, they differ fundamentally in their objectives and implementations.
>
> Our method, as outlined in the CReTIHC dataset, emphasizes designing causal reasoning tasks that mirror real-world scenarios. The core objective is for the Large Language Model (LLM) to critically assess causality, distinguishing genuine causal relationships from biases induced by language hallucinations and temporal events.
>
> Let's take an example :
>
> 1. **copper block was heated from fire.**
>
> 2. **pick the copper block by bare finger.**
>
> 3. **The skin burns.**
>
> Note: *The causal relationship between events 2 and 3 changes depending on which sentence is put between events 2 and 3.*
>
> Option 1 > **Enhanced : About 5 seconds have passed.**
>
> Option 2 > **Weaken : About 5 days have passed.**
>
> The former can strengthen the causal relationship, but the latter weakens the causal relationship because we know that the copper block cools down over the 5-day period.
>
> The key point lies in understanding how an new intervening event, occurring between two chronologically ordered events, impacts their existing causal relationship. This nuanced approach sets our method apart from adversarial training, which primarily confuses models by altering specific words to produce similar sentences.
>
> On the other hand, adversarial training typically involves generating perturbed or "adversarial" examples to challenge the model, ensuring it doesn't overfit to specific patterns in the training data and generalizes better to unseen data.
>
> In essence, while adversarial training seeks to improve model robustness against slight perturbations, our method aims to enhance a model's causal reasoning abilities by presenting it with complex scenarios that require a deeper understanding of cause-and-effect relationships.
>
> We hope this clarifies the distinction between our approach and adversarial training. We will ensure to make this differentiation more explicit in the revised manuscript to avoid any confusion.
>
> > R3: *'Improving commonsense causal reasoning by adversarial training and data augmentation" by I Staliunaite, PJ Gorinski, I Iacobacci (2021) have presented a very similar method of generating confounders as well as generating adversarial examples to improve commonsense causal reasoning models.’*
>
> **[Missing References]**
>
> We sincerely thank you for letting us know about valuable existing research. We recognize the importance of distinguishing our work from existing research. Our method focuses on constructing a new causal inference task using the e-CARE dataset, incorporating temporal interventions and hallucinated confoundings. Our approach requires discerning which sentence strengthens or weakens the causal relationship between two given sentences. In contrast, Staliunaite et al. [1] utilize adversarial training and data-augmentation techniques as a means to increase the data size for training causal inference models, given the limited size of the COPA dataset. We will clarify these distinctions in our revised manuscript and ensure that the mentioned paper is appropriately cited.
>
> [1] *Staliūnaitė, I., Gorinski, P. J., & Iacobacci, I. (2021, May). Improving commonsense causal reasoning by adversarial training and data augmentation. In Proceedings of the AAAI Conference on Artificial Intelligence (Vol. 35, No. 15, pp. 13834-13842).*
>
> > R4: *‘Line 204: Very repetitive explanation.’* & *‘Line 300: What does ‘reversely weaken’ mean? Is this meant to be described in contrast to strengthening? Better phrasing could help make it more clear.’*
>
> **[Typos and Presentation Improvements]**
>
> I saw what could be seen as repetitive explanations and unclear wording. It was intended to emphasize the meaning, but in the direction of the reviewer's advice, I will edit the mentioned line for clarity and conciseness. The term 'reversely weaken' is replaced with the explicit expression '(just) weaken' to avoid confusion.
>
> > R5: *‘Would be hard pressed to reproduce the results. The contribution depends on data that are simply not available outside the author's institution or consortium; not enough details are provided.’*
>
> **[Reproducibility Concerns]**
>
> All prompts necessary to generate and evaluate the data have been clarified. And we followed the standard procedure as described in the OpenAPI User Guide. For clarity and to ensure reproducibility, we will state all parameters, including default values, in the final version.
>
> We believe that our work makes a significant contribution to the field of causal inference in NLP. We hope that our explanations address your concerns, and we promise to make necessary corrections to improve the quality of the manuscript.
>
> Thank you for considering our paper.

---

### Official Review · Reviewer_NqUz · 2023-08-05

**Soundness:** 3

**Excitement:**

3: Ambivalent: It has merits (e.g., it reports state-of-the-art results, the idea is nice), but there are key weaknesses (e.g., it describes incremental work), and it can significantly benefit from another round of revision. However, I won't object to accepting it if my co-reviewers champion it.

**Paper Topic And Main Contributions:**

This paper presents CReTIHC, a novel dataset designed to test and enhance the causal reasoning abilities of LLMs.

**Reasons To Accept:**

1.  "Cause-and-effect" is a long and interesting research topic.

2.  The idea of redesigning existing natural language commonsense causal reasoning datasets will be promising in future Causal-aware tests.

**Reasons To Reject:**

1. Even for a short paper, this article lacks some necessary work, such as evaluating state-of-the-art models like GPT-4; at the same time, it also lacks necessary analysis of the results.

2. More in-depth analysis of the newly obtained data (beyond statistical data), including data distribution, comparison with other existing datasets, etc., has not been discussed.

**Reproducibility:**

3: Could reproduce the results with some difficulty. The settings of parameters are underspecified or subjectively determined; the training/evaluation data are not widely available.

**Reviewer Confidence:**

3: Pretty sure, but there's a chance I missed something. Although I have a good feel for this area in general, I did not carefully check the paper's details, e.g., the math, experimental design, or novelty.

---

> ### Author Rebuttal · Authors · 2023-08-29
>
> Thank you for your valuable comments.We appreciate the time and effort you've invested in reviewing our work. We would like to address the concerns you raised:
>
> > R1 : *‘1. Even for a short paper, this article lacks some necessary work, such as evaluating state-of-the-art models like GPT-4; at the same time, it also lacks necessary analysis of the results.’*
>
> **[Lack of Analysis with State-of-the-Art Models]**
>
> We understand the importance of evaluating datasets using state-of-the-art models such as GPT-4. During the internal review, there were concerns about **“circular evaluation”** because the datasets were created using the same model.
>
> However, based on your comments, we conducted further experiments with GPT-4. GPT-4 was tested on 100 samples like human evaluation, and showed slightly higher results than the results of human subjects with an accuracy of 0.92. We believe this is related to the "circular evaluation" concern.
>
> **[Insufficient Analysis of Results]**
>
> Our main focus is designing a new dataset to test the LLM's causal reasoning ability. The LLM's performance results on the data set presented in the "Experimental" section are intended to demonstrate the usefulness of the data set and new causal inference challenges rather than to provide an exhaustive analysis of causal understanding.
>
> We performed a human evaluation of 100 samples from the CReTIHC dataset in terms of result analysis. Three subjects were given the same prompts as in the LLM and evaluated causality between sentences. 100 samples were selected from the CReTIHC dataset, and one sample consists of 5 sentences. Subjects marked a sentence that strengthens the causal relationship between $\mathcal{S}1$ and $\mathcal{S}2$ as True, a sentence that weakens the causal relationship as False, and a sentence that does not change the causal relationship as None. All $\mathcal{A}1$ to $\mathcal{A}3$ were generated with one True/False/None target, but this information was not disclosed to the subjects. Therefore, the subjects solved three causal reasoning problems per sample, and the following figures were calculated from 300 test results.
>
> To provide a more detailed breakdown:
>
> |Evaluator|TRUE|FALSE|NONE|
> |--- |--- |--- |--- |
> |Human A|0.96|0.87|0.81|
> |Human B |0.92|0.91|0.89|
> |Human C |0.92|0.85|0.88|
>
>
> In this Table, TRUE refers to *Strengthened Causality*, FALSE indicates *Weakened Causality*, and NONE means *No Causality*.
>
> To assess inter-annotator agreements, we employed Fleiss' kappa. **The Fleiss' kappa coefficient was calculated to be 0.74**. Fleiss' kappa is a statistical measure used to determine the consistency of agreement between multiple raters. A kappa value above 0.7 typically indicates substantial agreement, suggesting that our human evaluators had a high level of consensus in their assessments. Only 2.3% of the sentences were rejected. Details including these results will be reflected in the final manuscript.
>
> > ‘R2: *More in-depth analysis of the newly obtained data (beyond statistical data), including data distribution, comparison with other existing datasets, etc., has not been discussed.’*
>
> **[Lack of In-depth Dataset Analysis]**
>
> Based on the E-CARE paper that started this study, we provided the same level of data set distribution information. Nonetheless, recognizing the importance of more in-depth data set analysis, the final version will include additional details such as basic statistics (word count, sentence length distribution) and examples in an appendix.
>
> We believe that our work makes a significant contribution to the field of causal reasoning in NLP. We hope that our clarifications address your concerns, and we are committed to making the necessary revisions to enhance the quality of our manuscript.
>
> Thank you for considering our paper.

---

### Official Review · Reviewer_EU9a · 2023-08-11

**Soundness:** 2

**Excitement:**

3: Ambivalent: It has merits (e.g., it reports state-of-the-art results, the idea is nice), but there are key weaknesses (e.g., it describes incremental work), and it can significantly benefit from another round of revision. However, I won't object to accepting it if my co-reviewers champion it.

**Paper Topic And Main Contributions:**

The paper presents a causal reasoning dataset with temporal interventions (a modification of e-CARE dataset). The paper further evaluated two models on the dataset and showed the models have limited accuracy.

**Reasons To Accept:**

- Models causal reasoning capabilities require more work and this paper adds to that literature and dataset.
- Making the current dataset more challenging to probe the models and provide opportunities to improve is a valuable contribution


**Reasons To Reject:**

- The modification of the e-CARE dataset is based on instruction to the language models. Though it’s a valid method, I do have concerns over the quality of the generated dataset (as acknowledged in the limitations section), and worry about the circular evaluation of the model generating the task and being evaluated on it.
- I would love to see at least a small section of sampling the generated data and doing some kind of human quality eval/control.


**Reproducibility:**

4: Could mostly reproduce the results, but there may be some variation because of sample variance or minor variations in their interpretation of the protocol or method.

**Reviewer Confidence:**

3: Pretty sure, but there's a chance I missed something. Although I have a good feel for this area in general, I did not carefully check the paper's details, e.g., the math, experimental design, or novelty.

---

> ### Author Rebuttal · Authors · 2023-08-29
>
> Thank you for your valuable comments. We appreciate the time and effort you've invested in reviewing our work. We would like to address the concerns you raised:
>
> > R1: *‘The modification of the e-CARE dataset is based on instruction to the language models. Though it’s a valid method, I do have concerns over the quality of the generated dataset (as acknowledged in the limitations section), and worry about the circular evaluation of the model generating the task and being evaluated on it.’*
>
> **[Concerns over Dataset Quality]**
>
> Our decision to employ LLMs for dataset generation is rooted in their ability to produce diverse and coherent sentences within specific guidelines, surpassing the constraints of solely relying on human annotators. This approach is supported by a technical report which emphasizes the efficiency of LLMs in data labeling, suggesting they can label data as efficiently as humans but at a faster pace [1].
>
> Furthermore, there are inherent advantages of LLM evaluation over human evaluation. As highlighted in a recent study, LLM evaluations are more reproducible. The challenges of human evaluations, such as the difficulty in hiring the same group of evaluators for repeated assessments, make them less consistent in comparison [2].
>
> To ensure quality, we implemented a self-feedback mechanism where the LLM iteratively assessed and refined its generated sentences. Only 44.1% of GPT-4 generated sentences met our quality criteria for inclusion in the CReTIHC dataset. Additionally, we conducted a human evaluation. (The results have been added below.)
>
> [1] *LLMs can label data as efficiently as humans, but at a faster pace (https://www.refuel.ai/blog-posts/llm-labeling-technical-report)*
>
> [2] *Chiang, C. H., & Lee, H. Y. Can Large Language Models Be an Alternative to Human Evaluations?. ACL 2023.*
>
> **[Circular Evaluation Concern]**
>
> Model Separation: As authors, we were interested in investigating whether there are Circular Evaluation Concerns when using the same model for data set creation and evaluation. Another study conducted by Wang et al. 2023 [3] confirmed the validity of our approach by simultaneously using one LLM for both generation and evaluation. Nonetheless, we have separated the model used for dataset generation (GPT-4) from the model used for evaluation (GPT-3.5). This further alleviates the circularity problem.
>
> Context-specific differences: Different contexts are used to generate and evaluate data, including prompts. By differentiating the prompt sentences used for data generation and evaluation, contextual conditions change within the LLM environment, similar to the difference between "CoT reasoning" and "without CoT reasoning."
>
> [3] *Wang, Y., Kordi, Y., Mishra, S., Liu, A., Smith, N. A., Khashabi, D., & Hajishirzi, H. Self-instruct: Aligning language model with self generated instructions. ACL 2023.*
>
> > R2: *‘I would love to see at least a small section of sampling the generated data and doing some kind of human quality eval/control.’*
>
> **[Human Evaluation]**
>
> We performed a human evaluation of 100 samples from the CReTIHC dataset. Three subjects were given the same prompts as in the LLM and evaluated causality between sentences. 100 samples were selected from the CReTIHC dataset, and one sample consists of 5 sentences. Subjects marked a sentence that strengthens the causal relationship between $\mathcal{S}1$ and $\mathcal{S}2$ as True, a sentence that weakens the causal relationship as False, and a sentence that does not change the causal relationship as None. All $\mathcal{A}1$ to $\mathcal{A}3$ were generated with one True/False/None target, but this information was not disclosed to the subjects. Therefore, the subjects solved three causal reasoning problems per sample, and the following figures were calculated from 300 test results.
>
> To provide a more detailed breakdown:
>
> |Evaluator|TRUE|FALSE|NONE|
> |--- |--- |--- |--- |
> |Human A|0.96|0.87|0.81|
> |Human B |0.92|0.91|0.89|
> |Human C |0.92|0.85|0.88|
>
> In this Table, TRUE refers to *Strengthened Causality*, FALSE indicates *Weakened Causality*, and NONE means *No Causality*.
>
> To assess inter-annotator agreements, we employed Fleiss' kappa. The **Fleiss' kappa coefficient was calculated to be 0.74**. Fleiss' kappa is a statistical measure used to determine the consistency of agreement between multiple raters. A kappa value above 0.7 typically indicates substantial agreement, suggesting that our human evaluators had a high level of consensus in their assessments. Only 2.3% of the sentences were rejected. Details including these results will be reflected in the final manuscript.
>
> We hope this format is more suitable for your needs.

---

### Meta-Review · Area_Chair_snod · 2023-09-18

**Recommendation:** 3

**Metareview:**

The work introduces a dataset for enhancing the causal reasoning of Large Language Models.

Among the major concerns expressed during the reviewing period, the lack of human evaluation of the contents produced by language models stands out (even though it is becoming a current practice supported by some investigations) and the circular evaluation of the model generating the task and being evaluated on it. The authors provided additional materials that tackle this issues and that should be included in the final version of the paper if accepted - together with all other comments.

---

### Decision · Program_Chairs · 2023-10-07

**Decision:**

Accept-Findings

**Comment:**

The work introduces a dataset for enhancing the causal reasoning of Large Language Models.

Among the major concerns expressed during the reviewing period, the lack of human evaluation of the contents produced by language models stands out (even though it is becoming a current practice supported by some investigations) and the circular evaluation of the model generating the task and being evaluated on it. The authors provided additional materials that tackle this issues and that should be included in the final version of the paper if accepted - together with all other comments.